# Studies in a Murine Granuloma Model of Instilled Carbon Nanotubes: Relevance to Sarcoidosis

**DOI:** 10.3390/ijms22073705

**Published:** 2021-04-02

**Authors:** Barbara P. Barna, Anagha Malur, Mary Jane Thomassen

**Affiliations:** Program in Lung Cell Biology and Translational Research, Division of Pulmonary, Critical Care and Sleep Medicine, Department of Medicine, Brody School of Medicine, East Carolina University, Greenville, NC 27834, USA; mutchka@oberlin.net (B.P.B.); malura@ecu.edu (A.M.)

**Keywords:** granuloma, sarcoidosis, multiwall carbon nanotubes, alveolar macrophages

## Abstract

Poorly soluble environmental antigens, including carbon pollutants, are thought to play a role in the incidence of human sarcoidosis, a chronic inflammatory granulomatous disease of unknown causation. Currently, engineered carbon products such as multiwall carbon nanotubes (MWCNT) are manufactured commercially and have been shown to elicit acute and chronic inflammatory responses in experimental animals, including the production of granulomas or fibrosis. Several years ago, we hypothesized that constructing an experimental model of chronic granulomatosis resembling that associated with sarcoidosis might be achieved by oropharyngeal instillation of MWCNT into mice. This review summarizes the results of our efforts to define mechanisms of granuloma formation and identify potential therapeutic targets for sarcoidosis. Evidence is presented linking findings from the murine MWCNT granuloma model to sarcoidosis pathophysiology. As our goal was to determine what pulmonary inflammatory pathways might be involved, we utilized mice of knock-out (KO) backgrounds which corresponded to deficiencies noted in sarcoidosis patients. A primary example of this approach was to study mice with a myeloid-specific knock-out of the lipid-regulated transcription factor, peroxisome proliferator-activated receptor gamma (PPARγ) which is strikingly depressed in sarcoidosis. Among the major findings associated with PPARγ KO mice compared to wild-type were: (1) exacerbation of granulomatous and fibrotic histopathology in response to MWCNT; (2) elevation of inflammatory mediators; and (3) pulmonary retention of a potentially antigenic ESAT-6 peptide co-instilled with MWCNT. In line with these data, we also observed that activation of PPARγ in wild-type mice by the PPARγ-specific ligand, rosiglitazone, significantly reduced both pulmonary granuloma and inflammatory mediator production. Similarly, recognition of a deficiency of ATP-binding cassette (ABC) lipid transporter ABCG1 in sarcoidosis led us to study MWCNT instillation in myeloid-specific ABCG1 KO mice. As anticipated, ABCG1 deficiency was associated with larger granulomas and increased levels of inflammatory mediators. Finally, a transcriptional survey of alveolar macrophages from MWCNT-instilled wild-type mice and human sarcoidosis patients revealed several common themes. One of the most prominent mediators identified in both human and mouse transcriptomic analyses was MMP12. Studies with MMP12 KO mice revealed similar acute reactions to those in wild-type but at chronic time points where wild-type maintained granulomatous disease, resolution occurred with MMP12 KO mice suggesting MMP12 is necessary for granuloma progression. In conclusion, these studies suggest that the MWCNT granuloma model has relevance to human sarcoidosis study, particularly with respect to immune-specific pathways.

## 1. Background

Sarcoidosis. Currently, no single definitive cause has been identified for sarcoidosis, an inflammatory granulomatous disease reported to have an incidence of 33.4 cases/year per 100,000 patients in the United States [1]. Diagnosis is established by clinical history, biopsy indicating non-caseating granulomas (which can be present in multiple organs), and negative studies for infectious agents [2]. Sarcoidosis is thought to occur as a reaction to poorly soluble environmental antigen(s) which have not yet been identified, with evidence that the disease may have multiple causes [3]. In approximately 90% of cases, sarcoidosis involves the lung, and up to 30% of patients may experience spontaneous remission [4]. Observation rather than treatment is favored in patients with minimal symptoms, good organ function, and no inflammation related to granulomas [4]. No biological markers have been established as assisting a definite diagnosis.

The rationale for the construction and study of a carbon nanotube-induced granuloma model of sarcoidosis was based upon numerous epidemiologic studies linking sarcoidosis incidence with environmental carbon exposures. Among the risk factors associated with sarcoidosis are exposures to wood-burning fireplaces or stoves and the occupation of firefighting [5,6]. Sarcoidosis has also been linked with occupational exposures to inorganic particulates and mycobacteria [5]. Dramatic evidence of relatively rapid elevation in sarcoidosis-like granulomatous disease was found in non-firefighting as well as fire-fighting individuals who were exposed to combustion materials released in the September 2001 World Trade Center (WTC) disaster [7,8,9]. After that catastrophic event caused by a plane striking the WTC building, follow-up studies of combustion products and some 20,000 responders revealed carbon nanotubes of diverse sizes in dust samples as well as within lung tissues of individuals affected with a pulmonary granulomatous disease resembling that of sarcoidosis [8]. Currently, carbon nanotube-based products are in use in several manufacturing fields such as textiles, polymer products, microelectronics, etc. [10]. Other applications of carbon nanotubes to pharmacy and medical products are under study [11].

## 2. Development of a Murine Multi-Wall Carbon Nanotube (MWCNT)-Induced Granuloma Model

The goal of our studies was to establish a chronic in vivo model of granulomatous inflammation mimicking sarcoid pathophysiology. In our model, MWCNT are administered via oropharyngeal instillation as a bolus into wild-type (C57/Bl6) mice. Initial dosage ranged from 25 to 100 ug with 100 ug providing consistent granuloma formation maintained up to 90 days after instillation [12]. When compared to an aerosol delivery method, retropharyngeal instillation may be superior as it achieves full and uniform delivery, with decreased variability between exposures [13,14]. A comparative study of lung tissue in inhalation versus instillation demonstrated smaller particle size and diminished pulmonary inflammation with inhalation [14]. However, investigations carried out to determine long-term effects of inhalation versus instillation with similar accumulated material burden had similar outcomes with both techniques [15,16]. Time course studies in our MWCNT model indicated well-formed granulomas as early as 10 days which persisted till 90 days. Detailed studies were performed at 60 days post-instillation. Gene and protein expression analyses were carried out on bronchoalveolar lavage (BAL) cells from MWCNT-instilled animals versus sham. Additional gene expression data from both granulomatous and non-granulomatous lung tissue were obtained by laser capture microdissection (LCM). Overall, results indicated significant pulmonary granulomatosis at 60 days, persisting up to 90 days with elevated BAL cell and lung tissue expression of granuloma-promoting factors, including interferon-gamma [17], osteopontin [18], matrix metalloproteinase-12 (MMP-12) [19], cell adhesion molecules [20], and elevated CD3+ T lymphocytes [12]. Differences were noted in gene expression levels within the granulomatous region of the lung as compared to the non-granulomatous parts of the lung. This model of an inhaled carbon nanotube-induced granulomatous disease was found to exhibit multiple similarities to sarcoidosis pathophysiology which are detailed in the following sections.

## 3. Role of PPARγ in Sarcoidosis

The transcription factor, peroxisome proliferator-activated receptor gamma (PPARγ), is part of a subfamily of nuclear receptors with ligand-inducible activity [21]. PPARγ is a critical regulator of lipid and glucose metabolism but also exhibits down-regulatory effects on genes linked to inflammatory events [22]. Alveolar macrophages from healthy individuals express constitutively high levels of PPARγ, in contrast to macrophages from other parts of the body [23]. Alveolar macrophages are the first line of defense in the lung which is constantly bombarded with environmental insults. PPARγ is usually present as a heterodimer complexed with retinoid X receptor alpha (RXRα) and bound to corepressors. With ligand stimulation, corepressor molecules are removed and the ligand, PPARγ, RXRα, and coactivators form an active complex, binding to PPARγ response elements (PPRE) [24]. Numerous ligands have been identified which can bind and activate PPARγ, leading to anti-inflammatory effects [24]. Ligand-bound PPARγ dampens the inflammatory response and is essential for orchestrating the lipid metabolism necessary for pulmonary surfactant processing and maintenance [25]. These activities are crucial for lung homeostasis. Interestingly, alveolar macrophages from macrophage-specific PPARγ KO mice, display increased inflammatory cytokine production, resulting in a Th1-like inflammatory response supporting the critical role of PPARγ in lung homeostasis [26]. Furthermore, analysis of the MWCNT granuloma model in wild-type mice also revealed decreased PPARγ activity and expression in alveolar macrophages from MWCNT-instilled mice after 60 days [27]. Disruption of PPARγ occurs in several human lung diseases including sarcoidosis [28]. It should be noted that in sarcoidosis, PPARγ deficiency in BAL cells is characteristic of severe inflammatory pulmonary disease associated with deteriorating pulmonary function which requires systemic treatment [29]. Mean PPARγ levels in sarcoidosis patients classified as having non-severe pulmonary disease (which has a high chance of spontaneous resolution) do not differ from those of healthy controls [29].

Interferon-gamma is a hallmark cytokine of sarcoidosis [30,31,32]. PPARγ and IFN-γ have demonstrated mutually antagonistic properties [29,33]. Furthermore, our previous in vitro studies with human alveolar macrophages demonstrated marked suppression of PPARγ by IFN-γ treatment [29]. Taken together these studies suggest that the mechanisms of PPARγ suppression may involve IFN-γ. Further support of this concept is provided by our studies in MMP12 KO mice where reduction of IFN-γ is accompanied by upregulation of PPARγ and granuloma resolution ([34], see Section 6. MMP12).

### 3.1. PPARγ Deficiency and Elevated Granulomatosis

In order to further define the role of PPARγ in granulomatous disease, we utilized macrophage-specific PPARγ KO mice in an attempt to better elucidate pathways associated with sarcoidosis. The most striking feature in PPARγ KO mice 60 days after oropharyngeal instillation of MWCNT was the presence of extensive granuloma formation within the lungs as assessed by histopathology and granuloma scoring indices compared to wild-type mice [27]. Furthermore, recruitment of increased CD3+ T cells to granulomatous foci in lungs was apparent by immunostaining [27]. Alveolar macrophages are the predominant cell type in BAL (93% PPARγ KO versus 97% wild-type). BAL lymphocyte counts were also higher in PPARγ KO mice (7%) than in wild-type (3%) and were not changed with MWCNT instillation [27]. Moreover, compared to wild-type controls, BAL cells from PPARγ KO control mice exhibited elevated expression of osteopontin and CCL2 (MCP-1) mediators, which were further increased by MWCNT instillation [27]. These findings indicated a tendency toward the elevation of inflammatory mediators in PPARγ KO animals and suggested that PPARγ might function as a negative regulator of granulomatous inflammation.

In a more recent study, we examined the effects of ligand-mediated activation of PPARγ in MWCNT-instilled wild-type mice [35]. Results indicated that activation of PPARγ via its specific ligand, rosiglitazone, inhibited granuloma formation, and reduced levels of the inflammatory mediators CCL2 and osteopontin via reduction of the pro-inflammatory transcription factor, NF-κB [35]. Both CCL2 and osteopontin expression are regulated by NF-κB activity [36,37]. These findings support the results of the previous study illustrating an exacerbation of inflammatory processes in PPARγ deficiency and significant negative regulatory effects of PPARγ on pulmonary granulomatous inflammation.

### 3.2. PPARγ Deficiency and Elevated Fibrosis

As fibrosis is present in some 20% of sarcoidosis patients and is a leading cause of mortality [38], we investigated the possibility of fibrotic changes in MWCNT-instilled PPARγ KO mice [39,40]. We also added another experimental group to the model by including 20 µg of ESAT-6, a 15-amino acid sequence of a mycobacterial antigen, in the instillation of MWCNT. Control groups received ESAT-6, MWCNT, or vehicle alone. The rationale for the ESAT-6 study was based upon human studies citing ESAT-6 as an inducer of T cell reactivity in sarcoidosis patients [41,42] as well as the suggestion that mycobacteria might constitute a causal element in the disease [43]. Moreover, the consequences of combining a microbial antigen within the MWCNT granuloma model had not been studied previously.

Initial experiments carried out in wild-type mice produced evidence that 60 days after concurrent instillation of the ESAT-6 peptide with MWCNT, both granulomatous involvement and pulmonary fibrosis had significantly increased compared to MWCNT instillation alone [39]. Interestingly, our previous studies in MWCNT-instilled wild-type mice had shown minimal evidence of fibrosis [12]. In ESAT-6/MWCNT instilled PPARγ KO mice, pathophysiology also increased in a similar fashion, with higher granuloma and fibrosis scores than with MWCNT alone [40]. Overall, granuloma and fibrosis pathology in ESAT-6/MWCNT instilled PPARγ KO mice exceeded values from wild-types [40]. The combination also elevated PPARγ KO fibronectin gene expression in granuloma-negative areas of instilled lungs compared to MWCNT alone. Interestingly, expression of the inflammatory mediators, osteopontin and CCL2, was not further enhanced in PPARγ KO BAL cells after instillation of combined MWCNT/ESAT-6. In PPARγ KO but not wild-type BAL fluids, however, significant elevations of fibrosis mediator proteins TGFβ, PDGFa, and IL-13 were found, suggesting that fibrotic pathways differed between wild-type and PPARγ KO mice [40].

Due to the differences in wild-type and PPARγ KO responses to ESAT-6 inclusion in the MWCNT model, pulmonary retention of ESAT-6 peptide in lung tissues was examined by mass spectrometry. Surprisingly, results indicated a lack of persistence of ESAT-6 in wild-type lungs at 60 days post instillation whereas ESAT-6 remained detectable in 60-day PPARγ KO lungs [40]. Such findings support previous data showing the capacity of PPARγ to enhance macrophage phagocytosis as part of a general resolution of inflammation [44]. Data further suggested that PPARγ deficiency might promote host reactivity to the antigenic ESAT-6 peptide via extended retention in tissue.

### 3.3. PPARγ Deficiency and Elevated Th-17 Lymphocytes

Our previous MWCNT instillation studies had noted elevated numbers of CD3+ T lymphocytes infiltrating pulmonary granulomatous areas in both wild-type and PPARγ KO mice [12,27]. Our most recent study explored the phenotypic characteristics of lymphocytes present in BAL fluids from PPARγ KO and wild-type mice instilled with control (vehicle alone), MWCNT, or MWCNT + ESAT-6 [45]. As noted previously, PPARγ KO mice exhibited increased BAL lymphocytes compared to wild-type [26] but also showed elevated CD4/CD8 ratios [45]. BAL lymphocytes from MWCNT + ESAT-6 instilled wild-type and PPARγ KO mice all displayed elevated expression of the T helper 1 (Th-1) transcription factor T-Bet, compared to controls. Unexpectedly, PPARγ KO BAL lymphocytes also exhibited increased markers of T Helper 17 (Th-17) cells compared to wild-type [45]. Current sarcoidosis studies show elevated Th-17 lymphocytes in both BAL fluids and peripheral blood from patients [32,46] and have proposed that Th-17 cells are necessary for promoting and reinforcing granuloma formation [47]. Such findings suggest links between PPARγ function, ESAT-6 stimulation, and T lymphocyte profiles which require further analyses in the MWCNT model.

## 4. ABCG1 Deficiency

Alveolar macrophage ATP-binding cassette (ABC) transporters are critical to surfactant clearance and pulmonary lipid regulation [48,49]. In human disease, transporter deficiencies have been associated with Tangier disease (ABCA1) [48] and pulmonary alveolar proteinosis (ABCG1) [50]. In sarcoidosis, we have reported that both ABCA1 and ABCG1 transporters are deficient in alveolar macrophages [51]. Moreover, ABCG1 is deficient in untreated PPARγ KO mice [52]. Further, we have noted that both ABCA1 and ABCG1 become decreased in both wild-type and PPARγ KO murine alveolar macrophages at 60 days after MWCNT instillation [51].

### 4.1. ABCG1 Deficiency and the MWCNT Granuloma Model

Based on these findings, a further study was carried out to determine the effects of ABC deficiencies in the MWCNT model [53]. Interestingly, results showed that deficiency of ABCA1 did not affect MWCNT-induced granuloma formation or pro-inflammatory gene expression. ABCG1 deficiency, however, clearly elevated pulmonary granulomatosis and fibrosis, as well as the expression of CCL2 and osteopontin mediators compared to wild-type mice [53]. These studies confirm the importance of intact PPARγ in regulating inflammatory disease as well as ABC transporter levels. Exacerbation of pulmonary granulomatous changes and inflammation were promoted by deficiencies of either PPARγ or transporter ABCG1.

### 4.2. ABCG1/ABCA1 Deficiencies and MicroRNA 33 Elevation in the MWCNT Model

Further studies of the MWCNT model in both wild-type and PPARγ KO mice revealed another regulatory pathway for ABC transporter control—that of microRNA 33 (miR-33) [51]. Mir-33 is a recognized regulator of ABCA1 and ABCG1 lipid transporters [53]. In MWCNT-instilled mice (either wild-type or PPARγ KO) elevated miR-33 was detectable in BAL cells [51]. Baseline miR-33 values were higher in sham PPARγ KO than wild-type, and MWCNT instillation elevated values in both strains. Concurrently, MWCNT instillation decreased levels of both ABCA1 and ABCG1 proteins in the lungs of both strains. In vitro studies confirmed that miR-33 overexpression induced decreases of both ABCA1 and ABCG1 gene expression in murine alveolar macrophages [51]. In vivo, elevated miR-33 expression was detected in granuloma but not non-granulomatous lung tissues in both wild-type and PPARγ KO mice. Murine data mirrored results from human sarcoidosis patients where elevated miR-33 gene expression was present in both BAL cells and granuloma tissues found in mediastinal lymph nodes compared to control groups [51]. ABCA1 and ABCG1 gene and protein expression were also significantly decreased in BAL cells from sarcoidosis patients compared to healthy controls [51]. In conclusion, the results suggest two possible pathways for transporter dysregulation in granulomatous disease: the first associated with intrinsic PPARγ status, and the second with miR33 up-regulation triggered by environmental challenges, such as MWCNT.

## 5. Common Gene Pathways in Sarcoidosis and the MWCNT Model

To assess the possibility of similar pathway mechanisms in granulomatous inflammation, murine BAL samples from MWCNT-instilled and sham control animals were simultaneously analyzed with human BAL samples from sarcoidosis patients and healthy controls. Transcriptional profiles from the mouse model were compared to those from human samples to identify overlapping molecular programs [17]. To better define activated pathways, integrated network and gene set enrichment analyses (GSEA) were performed. Application of GSEA to both murine and human samples uncovered an upregulation of overlapping gene sets in mice and sarcoidosis patients that represented approximately 40% of the total sets. To analyze the global response of murine cells to MWCNT, correspondence analysis, a form of multidimensional scaling, was applied to the entire murine microarray data set. Results showed a clear separation between control and MWCNT-instilled mice with a majority of differentially expressed genes involved primarily in immunity and inflammation. Commonly activated processes in both MWCNT mice and sarcoidosis patients included adaptive immunity, T-cell signaling, interleukin (IL)-12 and IL-17 pathways, IFN-γ signaling, apoptosis, and oxidative phosphorylation [17].

Quantitative PCR was used to validate four network genes identified as commonly upregulated in both the murine MWCNT model and in human sarcoidosis. Data confirmed elevated gene expression of MMP12, signal transducer and activator of transcription 4 (STAT4), cathepsin K (CTSK), and lymphocyte protein tyrosine kinase (LCK) [17]. These candidate genes are members of critical processes involved in lung remodeling and the immune response. For example, extracellular matrix-related gene products, including CTSK and MMP12, are critical to granuloma formation [34,54]. CTSK is one of a group of three related cysteine proteases which are present in the epithelioid and giant cells within granulomas [54]. Deficiencies of these proteases reduce the occurrence, composition, and formation of granulomas [54]. With respect to STAT4, activation is required in pathways leading to the expression of gamma interferon (IFN-γ) [55], a hallmark of inflammatory pathways in sarcoidosis [30].

## 6. MMP12 Deficiency and the MWCNT Granuloma Model

Our gene array studies on commonalities between the human sarcoidosis and murine MWCNT granuloma model showed that MMP12 was highly expressed in both species [17]. This suggested that MMP12 may be a key player for granuloma formation.

Initial studies of the MWCNT model noted a consistent elevation of the metalloproteinase, MMP12, in both granulomatous tissues and BAL cells of MWCNT-instilled mice [12]. MMP12 can be induced by the adhesion-promoting cytokine, osteopontin [56], which is also strongly elevated in MWCNT-instilled mice [12]. Previous studies have confirmed increased MMP12 gene and protein expression in sarcoidosis as well as the association of MMP12 elevation with disease severity [57]. To better understand the functions of MMP12, the MWCNT model was carried out in MMP-12 KO and wild-type mice [34]. Studies with MMP12 KO mice revealed acute reactions (10 days) similar to those of wild type. At chronic time points where wild-type maintained granulomatous disease (60 days), MMP12 KO mice exhibited attenuated granuloma formation as well as resolution, suggesting that MMP12 is necessary for granuloma progression. This coincided with elevated PPARγ and reduced IFN-γ expression in BAL cells, suggesting that these mediators may also be involved since previous studies have shown that PPARγ suppresses IFN-γ [29,33] and PPARγ deficiency amplifies granuloma formation [27]. These data strongly support a major role for MMP12 in granuloma formation and persistence.

## 7. Summary and Conclusions

A brief comparison of pulmonary mediator findings in the murine MWCNT model and in sarcoidosis patients indicates striking similarities between groups (Table 1). Among the elevated inflammatory mediators noted in both sarcoidosis and murine granuloma tissues were: osteopontin, MMP12, and the chemokine CCL2 (MCP-1) [12,57,58,59]. The presence of CD3+ T cells located in pulmonary granuloma foci from MWCNT-instilled mice was detected by immunostaining [12,27,40]. T cell accumulations in granulomatous tissues of sarcoidosis patients have also been consistently confirmed [47,60,61]. Moreover, comparative transcriptomics of MWCNT-instilled mice and sarcoidosis patients have indicated the presence of T cell signal transduction pathways within BAL cells [17] suggesting that, similar to sarcoidosis, the MWCNT model has an immune component. It should be noted again here that prominent deficiencies of the transcription factor, PPARγ, and ABC transporters ABCA1 and ABCG1 have also been found in both the MWCNT model and sarcoidosis (Table 1, Figure 1).

As noted in a recent review of in vitro/in vivo sarcoidosis models, more innovative approaches are needed to explore the pathogenesis and pathophysiology of sarcoidosis [67]. A focus on granuloma formation alone is insufficient to provide information on mechanisms of fibrosis formation which affects some 20% of sarcoidosis patients [61]. Interestingly, the MWCNT model illustrates some fibrotic changes along with those relating to granuloma development but more so in some of the knock-out mouse strains, which may provide clues to pathogenic pathways. The representative histology of each of the mouse strains is shown in Figure 2. Wild-type (C57Bl/6) histology shows granuloma formation at 60 days with minimal fibrosis. PPARγ KO and ABCG1 KO demonstrate prominent granulomatous and fibrotic inflammation. MMP 12 KO show markedly attenuated granuloma formation and little fibrosis. Collectively, findings from the above MWCNT studies support usage of the MWCNT model as biologically relevant for sarcoidosis studies. The structure of the model is unique in that it provides a chronic inflammatory milieu lasting for 90 days. Comparative analyses have uncovered substantial pathway similarities between the MWCNT model and human sarcoidosis pathophysiology. Such results suggest that future exploration of novel therapeutics in sarcoidosis patients may benefit from utilizing the murine MWCNT model.

## Figures and Tables

**Figure 1 ijms-22-03705-f001:**
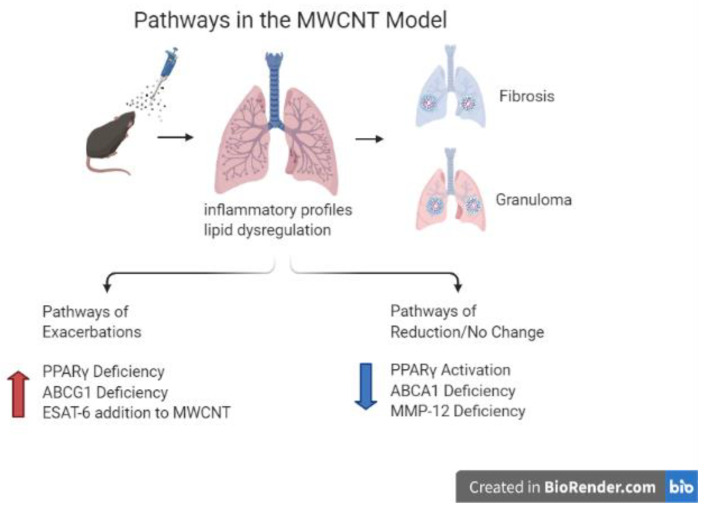
Schematic representation of proposed mechanisms affecting MWCNT-induced chronic granulomatous disease and fibrosis in mice. Mice instilled with MWCNT have an inflammatory response and lipid dysregulation. These lungs develop granulomas and depending on the challenge and strain may become fibrotic. Exacerbations are caused by deficiencies in PPARγ, ABCG1 and addition of ESAT-6. Attenuation occurs with increased PPARγ activation, ABCA1 deficiency has no effect and MMP-12 deficiency results in reduction of granulomas.

**Figure 2 ijms-22-03705-f002:**
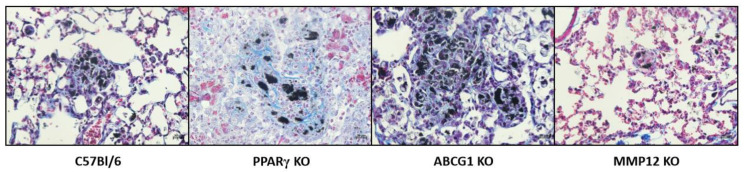
Representative trichrome staining from mice instilled with MWCNT for 60 days. Magnification 400X. C57Bl/6, ABCG1 KO, and MMP12 KO stained with Gomori’s trichrome. PPARγ KO stained with Masson’s trichrome.

**Table 1 ijms-22-03705-t001:** Findings common to the MWCNT model and sarcoidosis.

Mediators and Status	Role in Pathophysiology	Pulmonary Locations	
		MWCNT MODEL	SARCOIDOSIS
Peroxisome proliferator-activated receptor gamma (PPARγ)—Deficiency	Granuloma Formation [27]	Bronchoalveolar lavage (BAL) cells [27]	BAL cells [28]
	Lipid Dysregulation [22]		
	Fibrosis [40]		
	Inflammatory Profiles [26]		
Osteopontin—Elevation	Granuloma Formation [18]	Granuloma tissue, BAL cells, BAL fluids [12,62]	Granuloma tissue [58]
Matrix-metalloproteinase 12 (MMP-12)—Elevation	Granuloma Formation [34]	Granuloma tissue, BAL cells (11) [34]	Granuloma tissue, BAL cells [57]
CCL2 (MCP-1)—Elevation	Granuloma Formation [63]	Granuloma tissue [12,62]	Granuloma tissue [59]
ABCG1—Deficiency	Granuloma Formation [53]	BAL cells [51]	BAL cells [51]
	Fibrosis [53]		
	Lipid Dysregulation [51]		
Tumor Necrosis Factor alpha (TNFα)—Elevation	Macrophage M1 Inflammatory Profile [64]	Granuloma tissue (11)	BAL cells [64]
Interferon-gamma (IFN-γ)—Elevation	T Cell Inflammatory Profile [65]	Granuloma tissue, BAL cells [17]	BAL cells [30]
Signal Transducer and Activator of Transcription (STAT 4)—Elevation	T Cell Inflammatory Profile [55]	BAL cells [17]	BAL cells [17]
Cathepsin K (CTSK)—Elevation	Granuloma Formation [54]	BAL cells [17]	BAL cells [17]
TWIST 1—Elevation	Macrophage M1 Inflammatory Profile [60]	BAL cells [60]	BAL cells [60]
Th17 cells—Elevation	Inflammatory Profile [32]	BAL cells [45]	BAL cells [32]
	Granuloma Formation [47]		
T cells—Elevation	Inflammatory Profile [12,26]	Granuloma tissue [27,40,45]	Granuloma tissue [47]
ABCA1—Deficiency	Lipid Dysregulation [48]	BAL cells [51]	BAL cells [51]
MicroRNA 33 (Mir-33)—Elevation	Lipid Dysregulation [66]	Granuloma tissue, BAL cells [51]	Granuloma tissue, BAL cells [51]

## Data Availability

All data is publically available.

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
