# Peer review of "Studies in a Murine Granuloma Model of Instilled Carbon Nanotubes: Relevance to Sarcoidosis"

_ijms, 2021, doi:10.3390/ijms22073705_

Round 1
Reviewer 1 Report
The authors in this manuscript describing a granuloma mouse model in response to carbon nanotube that they developed in their group. In multiple studies they showed that murine granuloma model in the background of PPARγ KO exhibit larger granuloma size and inflammatory cytokines that have some resembles to sarcoidosis pathology. While the murine model might have some merit, the authors failed to discuss real human sarcoidosis pathology. Furthermore, it appears that authors ignore large body of relevant literature, while reaping their murine model finding to a great extent. For example section 3.1. PPARγ deficiency and Elevated Granulomatosis: “Recruitment of CD3+ T cells to granulomatous foci in the lungs was apparent by immunostaining. As anticipated from previous studies…”
Despite extensive explanation of MWCNT Model, the bridge to human disease is poorly developed. The section 5. Is poorly developed.
Most importantly, it is less likely that all sarcoidosis patients have a loss of PPARγ function. How they reconcile spontaneous resolution of disease if all these patients have a loss of PPARγ function?
It would be helpful to the sarcoidosis field to know how the signaling of PPARγ is regulated.
Or what signaling might downregulate PPARγ in these patients?
What is the role of cathepsin k in this disease? Authors provide many dots, which are poorly connected and they are not giving the whole picture.
Furthermore, having a figure to describe their model will help the readers .
Author Response
Reviewer #1
(1). The authors in this manuscript describing a granuloma mouse model in response to carbon nanotube that they developed in their group. In multiple studies they showed that murine granuloma model in the background of PPARγ KO exhibit larger granuloma size and inflammatory cytokines that have some resembles to sarcoidosis pathology. While the murine model might have some merit, the authors failed to discuss real human sarcoidosis pathology. Furthermore, it appears that authors ignore large body of relevant literature, while reaping their murine model finding to a great extent. For example section 3.1. PPARγ deficiency and Elevated Granulomatosis: “Recruitment of CD3+ T cells to granulomatous foci in the lungs was apparent by immunostaining. As anticipated from previous studies…”
Reply. Additional description of sarcoidosis pathology and a recent review reference have been added to Section # 1 (Background) in the Sarcoidosis section. As the reviewer suggests, the sentences in Section # 3.1 that mention CD3+ T cells in PPARγ KO mice are confusing and have been revised to better clarify lymphocyte and macrophage cell counts.
(2). Despite extensive explanation of MWCNT Model, the bridge to human disease is poorly developed. The section 5. Is poorly developed.
Reply. Section # 5 focuses on identification of common gene pathways by transcriptomics and subsequent validation by quantitative PCR in BAL cells from pulmonary granuloma-bearing sarcoidosis patients and mice. We have also changed the title of this section to “Common Gene Pathways in Sarcoidosis and the MWCNT Model” to more accurately reflect the contents of this paragraph.
(3). Most importantly, it is less likely that all sarcoidosis patients have a loss of PPARγ function. How they reconcile spontaneous resolution of disease if all these patients have a loss of PPARγ function?
Reply. We have added data to Section #3 to clarify the question of PPARγ function vs spontaneous resolution of disease. Not all sarcoidosis patients lose PPARγ function. Patients diagnosed with non-severe disease are likely to experience spontaneous resolution, are not treated, and have PPARγ levels which do not significantly differ from those of healthy controls. See Reference # 29.
(4). It would be helpful to the sarcoidosis field to know how the signaling of PPARγ is regulated.
Reply. In Section 3, we have added data describing the complex activation of PPARγ by a variety of ligands in the promotion of anti-inflammatory effects.
(5). Or what signaling might downregulate PPARγ in these patients?
Reply. We have also added some information about interferon gamma regulation of PPARγ activity. “Interferon gamma is a hallmark cytokine of sarcoidosis (30-32). PPARγ and IFN-γ have demonstrated mutually antagonistic properties (29, 33). Furthermore, our previous in vitro studies with human alveolar macrophages demonstrated marked suppression of PPARγ by IFN-γ treatment (29). Taken together these studies suggest that the mechanisms of PPARγ suppression may involve IFN-γ. Further support of this concept is provided by our studies in MMP12 KO mice where reduction of IFN-γ is accompanied by upregulation of PPARγ and granuloma resolution [(34), see section 6. MMP12].”
(6). What is the role of cathepsin k in this disease? Authors provide many dots, which are poorly connected, and they are not giving the whole picture.
Reply. Additional information on cathepsin K has been provided in Section 5.
(7). Furthermore, having a figure to describe their model will help the readers.
Reply. We have now added figure 1 to represent the model.
Reviewer 2 Report
In the current manuscript, the authors summarize their and other groups' findings regarding the induction of granulomas as a model of sarcoidosis in knock-out mice. The paper is comprehensive, yet well-structured and an excellent read. I have a few suggestions for improvement:
- In line 74 it is mentioned that granuloma formation is 'maintained up to 90 days after instillation'. What happens after 90 days? Do the granulomas disappear / alter their appearance? Are the mice always sacrificed at 90- days at the latest? Please elaborate.
- Although it is explained in a later section, it may be worth a short comment in line 119-120 as to which cell types are seen to be increased in BAL.
- The paper could benefit from a figure with a few representative micrographs and perhaps also immunohistochemical stains of the granulomas induced in the various model which are discussed.
Author Response
Reviewer #2
In the current manuscript, the authors summarize their and other groups' findings regarding the induction of granulomas as a model of sarcoidosis in knock-out mice. The paper is comprehensive, yet well-structured and an excellent read. I have a few suggestions for improvement:
- In line 74 it is mentioned that granuloma formation is 'maintained up to 90 days after instillation'. What happens after 90 days? Do the granulomas disappear / alter their appearance? Are the mice always sacrificed at 90- days at the latest? Please elaborate.
REPLY: The granulomas are still present at 120 days post-instillation in wild-type mice. Most of our studies have been carried out at 60 days primarily for economical reasons. We do not have extensive data on the 120 days other than some histology with a few pilot animals.
- Although it is explained in a later section, it may be worth a short comment in line 119-120 as to which cell types are seen to be increased in BAL.
REPLY: A statement on cell types identified in PPARγ-KO and wild-type mice has been inserted into Section 3.1.
- The paper could benefit from a figure with a few representative micrographs and perhaps also immunohistochemical stains of the granulomas induced in the various model which are discussed.
REPLY: We have included a diagram of the proposed model (figure1), and trichrome stain of the various models discussed (figure 2).